# Vitamin B12 Status Upon Short-Term Intervention with a Vegan Diet—A Randomized Controlled Trial in Healthy Participants

**DOI:** 10.3390/nu11112815

**Published:** 2019-11-18

**Authors:** Ann-Kathrin Lederer, Luciana Hannibal, Manuel Hettich, Sidney Behringer, Ute Spiekerkoetter, Carmen Steinborn, Carsten Gründemann, Amy Marisa Zimmermann-Klemd, Alexander Müller, Thomas Simmet, Michael Schmiech, Andrea Maul-Pavicic, Yvonne Samstag, Roman Huber

**Affiliations:** 1Center for Complementary Medicine, Institute for Infection Prevention and Hospital Epidemiology, Medical Center—University of Freiburg, Faculty of Medicine, University of Freiburg, 79106 Freiburg, Germany; manuel.hettich@uniklinik-freiburg.de (M.H.); carmen.steinborn@uniklinik-freiburg.de (C.S.); carsten.gruendemann@uniklinik-freiburg.de (C.G.); amy.klemd@uniklinik-freiburg.de (A.M.Z.-K.); alexander.mueller@uniklinik-freiburg.de (A.M.); roman.huber@uniklinik-freiburg.de (R.H.); 2Laboratory of Clinical Biochemistry and Metabolism, Department of General Pediatrics, Medical Center—University of Freiburg, Adolescent Medicine and Neonatology, Faculty of Medicine, University of Freiburg, 79106 Freiburg, Germany; luciana.hannibal@uniklinik-freiburg.de (L.H.); sidney.behringer@uniklinik-freiburg.de (S.B.); ute.spiekerkoetter@uniklinik-freiburg.de (U.S.); 3Institute of Pharmacology of Natural Products & Clinical Pharmacology, Ulm University, 89081 Ulm, Germany; thomas.simmet@uni-ulm.de (T.S.); michael.schmiech@uni-ulm.de (M.S.); 4Department of Rheumatology and Clinical Immunology, Medical Center—University of Freiburg, Faculty of Medicine, University of Freiburg, 79106 Freiburg, Germany; andrea.maul-pavicic@uniklinik-freiburg.de; 5Center of Chronic Immunodeficiency (CCI), Medical Center—University of Freiburg, Faculty of Medicine, University of Freiburg, 79106 Freiburg, Germany; 6Institute of Immunology, Section Molecular Immunology, University of Heidelberg, 69120 Heidelberg, Germany; yvonne.samstag@urz.uni-heidelberg.de

**Keywords:** vegan nutrition, holotranscobalamin, fatty acids, vitamin B_12_, micronutrient

## Abstract

Vegans are at an increased risk for certain micronutrient deficiencies, foremost of vitamin B_12_. Little is known about the short-term effects of dietary change to plant-based nutrition on vitamin B_12_ metabolism. Systemic biomarkers of vitamin B_12_ status, namely, serum vitamin B_12_ and holotranscobalamin, may respond quickly to a reduced intake of vitamin B_12_. To test this hypothesis, 53 healthy omnivore subjects were randomized to a controlled unsupplemented vegan diet (VD, *n* = 26) or meat-rich diet (MD, *n* = 27) for 4 weeks. Vitamin B_12_ status was examined by measurement of serum vitamin B_12_, holotranscobalamin (holo-TC), methylmalonic acid (MMA) and total plasma homocysteine (tHcy). Holo-TC decreased significantly in the VD compared to the MD group after four weeks of intervention, whereas metabolites MMA and tHcy were unaffected. Body weight remained stable in both groups. VD intervention led to a significant reduction of cholesterol intake, and adequate profiles of nutrient and micronutrient status. Lower intake of vitamin B_12_ was observed in VD, which was mirrored by a lower concentration of serum vitamin B_12_ and reduced holo-TC after 4 weeks. Plasma holo-TC may be a fast-responding biomarker to monitor adequate supply of vitamin B_12_ in plant-based individuals.

## 1. Introduction

Veganism is increasingly popular in Western societies [1,2]. Due to mostly health-, ecological- and/or ethical reasons, vegans exclude all animal products from their diet and are therefore at an increased risk for certain micronutrient deficiencies [3,4]. Particularly relevant is vitamin B_12_, as this micronutrient is exclusively synthesized by a few groups of bacteria and archaea; very few plant sources carry over traces of vitamin B_12_ from microbial metabolism [4,5]. Vitamin B_12_, also called cobalamin, is involved in the metabolism of all cells and is crucial for DNA synthesis [4,6]. A deficiency of vitamin B_12_ leads to megaloblastic anemia and peripheral neuropathy [4]. Vitamin B_12_ is a coenzyme for two enzyme-catalyzed reactions in the human body. Firstly, the remethylation of methionine from homocysteine is catalyzed by methionine synthase and requires methylcobalamin [4,7]. Deficiency of vitamin B_12_ results in buildup of homocysteine (Hcy), a decrease of the essential amino acid methionine (Met) and impaired regeneration of tetrahydrofolate from 5-methyltetrahydrofolate via the inactivation of methionine synthase. Secondly, the isomerization of methylmalonyl-CoA to succinyl-CoA requires adenosylcobalamin in a reaction that replenishes succinyl-CoA into the tricarboxylic acid cycle in energy metabolism [6]. Methylmalonyl-CoA mutase is part of amino acid metabolism [7]. Deficiency of vitamin B_12_ leads to an accumulation of methylmalonic acid (MMA) due to inactivation of mitochondrial methylmalonyl-CoA mutase. Elevated concentrations of Hcy and MMA are toxic; therefore, cells export these metabolites into circulation (Figure 1). Thus, plasma Hcy and MMA derive exclusively from intracellular enzymatic reactions and are often referred to as cellular markers of vitamin B_12_ status. The transport of dietary vitamin B_12_ requires a complex relay of the micronutrient via three transport proteins that bind sequentially to vitamin B_12_, namely haptocorrin (saliva and stomach), intrinsic factor (stomach and intestine), and transcobalamin (intestine into circulation) [8]. All dietary vitamin B_12_ is ultimately transferred to transport protein transcobalamin (TC). TC binds vitamin B_12_ after intestinal absorption and carries it to the liver where it reaches systemic circulation for the delivery of the micronutrient to all cells in the body [8,9] (Figure 1). The vitamin B_12_-bound form of transcobalamin is known as holotranscobalamin (holo-TC), a widely used systemic biomarker of vitamin B_12_ status [6,10,11,12,13]. The use of a combination of biomarkers is recommended for the reliable assessment of vitamin B_12_ status, including serum vitamin B_12_, holo-TC, MMA and total plasma Hcy (tHcy) [4,6].

Long-term, vegan diets with insufficient intake or without supplementation of vitamin B_12_ can lead to vitamin B_12_ deficiency [6]. Two studies showed high prevalence of low holo-TC in life-long vegetarian Indians [14,15]. One study showed lower plasma concentration of holo-TC in vegetarian Indians with coronary artery disease compared to their healthy control counterparts [14]. Studies performed in humans who underwent total gastrectomy due to cancer or other reasons, hence eliminating intrinsic factor biosynthesis, showed that clinical consequences of vitamin B_12_ deficiency appear 2 to 5 years after surgery [16]. Vitamin B_12_ is mainly stored in liver and kidney [17]. It has been estimated that daily losses of vitamin B_12_ are normally less than 0.1% of total body stores in adults [16]. It has been proposed that the onset of vitamin B_12_ deficiency in unsupplemented vegetarians may take even a longer time to manifest since in these individuals the absorption machinery is intact [18]. The possibility that markers of vitamin B_12_ status such as holo-TC may decrease upon short-term limited intake of vitamin B_12_ without an evident deficiency of the micronutrient has been considered [16]. However, the early events that occur upon introduction of a vitamin B_12_-deficient diet on biomarkers of vitamin B_12_ metabolism have not been studied thus far. Likewise, the short-term effects of plant-based diets on plasma markers of inflammation and cardiovascular health have not been examined. This study was designed to test two hypotheses, namely: (a) that systemic biomarkers of vitamin B_12_ status, that is, serum vitamin B_12_ and its biological active form, holotranscobalamin, may respond quickly to a reduction in the intake of vitamin B_12_, and (b) that markers of inflammation and cardiovascular health may be modified by the presence or absence of animal products in the diet. Herein, we report results on biomarkers of vitamin B_12_ status, plasma and urine markers relevant to inflammation and cardiovascular health, and a comprehensive nutritional assessment of vegan diet (VD) and meat diet (MD) groups.

## 2. Materials and Methods

The monocentric, controlled, randomized, explorative trial with parallel groups was performed between April and June 2017. It was registered at the German Clinical Trial register (DRKS00011963) and approved by the ethical committee of the University Medical Center of Freiburg, Germany (EK Freiburg 38/17) before onset. The study was performed according to the principles of the declaration of Helsinki and to the guidelines of ICH (International Conference on Harmonization) for good clinical practice (GCP). All subjects provided written informed consent before participation. All data were handled with strict confidentiality.

### 2.1. Course of the Study

Participants were recruited via newspaper announcement. Interested persons were screened for eligibility by phone call and invited for a personal examination. Individuals with confirmed eligibility were included in the study after providing written informed consent. Subjects were instructed on how to prepare their own balanced mixed diet according to the guidelines of the German Nutrition Association (DGE) [20]. BMI was measured and subjects had to stay on a balanced mixed diet for one week. After this one-week run-in phase baseline blood and urine samples were taken early in the morning. Subjects were randomly assigned to either a meat-rich (>150 g of meat daily) or a strict vegan diet for four weeks. Each subject received extensive training on their assigned diet. In addition, participants received detailed information material, a recipe book (see Appendix A) and one free breakfast. The subjects were in contact with the study staff on at least a weekly basis by phone or email. During the trial, all subjects had the opportunity to eat for free at the restaurant of the University Hospital Freiburg, where a customized vegan meal was offered for the participants of the study in addition to the regular predominantly meat-based buffet menu. In addition, some local restaurants offered discounts for study participants. After four weeks, fasting blood and urine samples were taken and the trial was finished. All subjects recorded their food intake in a standardized nutritional protocol (“Freiburg nutritional protocol”, see Appendix A) for the run-in phase and the entire trial. The protocol summarizes different foods to 15 superordinate categories, and nutritional intake has to be documented as serving, which consists of a pre-defined amount (g) or an absolute amount (g). All subjects were asked to keep their body weight stable (daily weighing) and to not implement changes in their lifestyle.

Nutritional protocols were evaluated via NutriGuide^®^ (Version 4.7, Nutri-Science GmbH, Hausach, Germany). Only protocols with a plausible energy intake of more than 1100 kcal daily were considered. All results were collected in a predesigned table. The data were entered and analyzed blinded for diet assignation. Blood and urine sample identification was performed after completion of all examinations and analyses.

### 2.2. Inclusion and Exclusion Criteria

Healthy subjects between 18 and 60 years of age with a Body Mass Index (BMI) between 21 kg/m^2^ and 30 kg/m^2^ were considered eligible for inclusion in the study. Criteria of exclusion were the intake of a long-term medication, pregnancy or lactation, severe allergies, a pre-existing vegan or vegetarian diet as well as an aversion to vegan diet or animal products, an eating disorder, participation in another clinical trial in the last 4 weeks, blood donation ≤4 weeks before trial onset and self-reported drug, alcohol (>20 g/day), or daily nicotine abuse. Eligible persons had to be able to speak and understand German and be willing to complete a nutritional protocol.

### 2.3. Randomization and Blinding

Randomization was carried out after the one week run in phase directly after the first blood samples were taken. The randomization list was created electronically by a third independent person (Python Software, Python Software Foundation, Wilmington, DE, USA). Blinding was not possible due to the nature of the intervention.

### 2.4. Selection of Target Parameters

The study aimed to answer two major questions: (a) Do systemic markers of vitamin B_12_ status respond to short-term dietary intervention with a meat-rich or a plant-based diet? and (b) Do meat-rich and vegan diets have an impact on plasma markers of inflammation and cardiovascular disease?

The primary target was the status of vitamin B_12_ (measured by serum vitamin B_12_, holo-TC and MMA) at the end of the trial (question a). Aiming to evaluate indicators of cellular vitamin B_12_ metabolism, measurement was complemented by serum concentration of Met and tHcy.

Vitamin B_12_, folate (vitamin B_9_) and vitamin B_6_ are the three major nutritional modifiers of homocysteine concentration in humans [21]. We therefore examined folate and vitamin B_6_ intake in the respective diet groups and measured metabolites in pathways that require Hcy homeostasis, i.e., methionine cycle B_12_- and B_9_-dependent (Met) and the trans-sulfuration pathway, B_6_-dependent (cysteine, cystathionine) and downstream product glutathione (GSH). Methionine sulfoxide (MSO) was measured as a means for assessing quality of sample (i.e., oxidation); no changes in MSO are expected in healthy subjects [22].

Regarding fatty acids, the targets were the serum concentrations of arachidonic acid, docosahexaenoic acid, eicosanoic acid, eicosenoic acid, linoleic acid, linolenic acid, and oleic acid. Additionally, urine concentrations of prostaglandin E2, which is one of the final metabolic products of arachidonic acid, were measured.

Another crucial micronutrient in VD is vitamin D, which was measured by 25-OH-vitamin D_2_/D_3_. Due to an expected higher intake of nitrate in those consuming a vegan diet the serum concentration of nitrate and nitrite was measured and related to the results of the nutritional protocols.

### 2.5. Blood and Urine Analysis

Cryovials for serum and urine were labeled without information about diet group. External laboratories did not receive information concerning patient randomization. Serum from each participant was collected and aliquoted in six separate 1.5-mL cryovials and stored at −20 °C until measurement or transportation to external laboratories. Urine samples were aliquoted in six 1 mL cryovials and were frozen immediately at −20 °C until measurement or transportation to external laboratories. Dry ice was used for transport of samples to external laboratories. All methods were previously established and validated. Measurement of holo-TC and 25-OH-vitamin D_2_/D_3_ were performed by the Central Laboratory of the University Medical Center of Freiburg. Measurement of serum vitamin B_12_ was performed by Labor Enders, Stuttgart.

### 2.6. Analysis of Serum tHcy, Cys, Cystathionine, GSH, Met, and MSO by LC-ESI-MS/MS

Total levels of homocysteine, cysteine, cystathionine, glutathione, methionine, and methionine sulfoxide were measured by LC-MS/MS according to a published procedure [23]. Briefly, 20 µL of plasma was mixed with 20 µL internal standard (*S*-methylglutathione, Sigma-Aldrich Chemie GmbH, Taufkirchen, Germany) and 20 µL 0.5 M dithiothreitol (DTT, Sigma-Aldrich Chemie GmbH, Taufkirchen, Germany). The samples were vortexed and incubated at room temperature for 15 minutes. Proteins were precipitated by addition of 100 µL 0.1% formic acid in methanol. Samples were centrifuged at 13,000 rpm for 5 min and 80 µL of supernatant was transferred to HPLC vials. Samples were measured after injecting 2 µL of sample into the LC-ESI-MS/MS machine (Sciex QTrap 6500+, Sciex, Darmstadt Germany). Quantification was performed via calibration curves for each one of the metabolites of interest, using pure commercial standards. Single measurements were performed for each sample. Assay performance quality was examined by incorporating a commercially available standardized marker for Hcy in plasma (Control special assays in serum, MCA, Product Nrs. SAS-02.1 and SAS-02.2).

### 2.7. Analysis of Methylmalonic Acid (MMA) by LC-ESI-MS/MS (Sciex QTrap 6500+, Sciex)

Methylmalonic acid was determined based on a previously published method with modifications [24]. The assay was optimized to employ 50 µL of plasma. Single measurements were performed for each sample. Assay performance quality was examined by incorporating a commercially available standardized marker for MMA in plasma (Control special assays in serum, MCA, Product Nrs. SAS-02.1 and SAS-02.2).

### 2.8. Amino Acids Profiles

Amino acid profiles in serum were measured after precipitating proteins by adding 50 µL of 10% aqueous sulfosalicylic acid to 200 µL of serum. Samples were centrifuged for 8 min at 3600 *g* and the supernatant was analyzed with a Biochrom 30 amino analyzer (Biochrom Ltd., Cambridge, UK), using ion exchange chromatography and post-column ninhydrine derivatization. Single measurements were performed for each sample. Quality controls from ClinChek were included in each assay (ClinCheck, Level I and Level II, plasma control, lyophilized for amino acids (AAA), product Nrs. 10280 and 10281, Recipe^®^, GmbH, Munich, Germany).

### 2.9. Fatty Acid Profiles

Fatty acids profiles in serum were measured as methyl esters of the respective carboxylates by GC-MS (gas chromatography–mass spectrometry). Sample preparation was performed by mixing 200 µL plasma with 200 µL of internal standard solution (50 µM D4-behenic acid and 50 µM D4-lignoceric acid in chloroform/methanol (2 parts chloroform and 1 part methanol). 1 mL of an acetonitrile/hydrochloric acid solution (4 parts acetonitrile and 1 part 37% aqueous hydrochloric acid) was added and the sample was vortexed. The sample was then incubated for 2 h at 90 °C. Samples were cooled down to room temperature and the fatty acids were extracted with 2 mL n-hexane. The hexane-phase was transferred into another glass vial and hexane was evaporated at 55 °C with a gentle nitrogen flow. The remaining fatty acids were methylated with freshly prepared diazomethane (this converts carboxylates into the corresponding methyl-esters) and the excess of the derivatizing agent was eliminated using gaseous CO_2_. The residue was dissolved in 200 µL hexane and analyzed by GC-MS. Fatty acids of interest were quantified relying on the two stable isotopically labeled internal standards d4-C22 ((3,3,5,5-2H4) docosanoic acid and d4-C24 ((3,3,5,5-2H4) tetracosanoic acid). Single measurements were performed for each sample, with the quantifications performed versus two isotopically labeled standards. Quality controls were included in each assay (Control special assays in serum, MCA, Product Nrs. SAS-02.1 and SAS-02.2).

### 2.10. Analysis of Total Nitrite and Nitrate

The total concentration of nitrite and nitrate in serum was determined with a colorimetric assay based on the Griess reaction purchased from Cayman Chemicals (Product Nr. 780001). Calibrators were measured in duplicates. Single measurements were performed for each sample.

### 2.11. Urine Analysis

Quantification of tetranor-PGEM (tetranor-prostaglandin E metabolite) was performed based on the published procedure by Idborg et al. [25]. The method was optimized and validated in terms of linearity, precision, sensitivity, selectivity, accuracy, recovery, limit of detection, and limit of quantification based on the standardization criteria of DIN 32645 as defined by the German standardization committee. The tetranor-PGEM concentrations were standardized by the individual creatinine concentrations quantified in urine samples. Analysis of creatinine was carried out by the department of Clinical Chemistry and Analytics (University Hospital Ulm) by a modified Jaffe reaction (CREP2, Roche Diagnostics, Rotkreuz, Switzerland).

### 2.12. Statistical Analysis

Sample size was planned for a pilot study, because effect size was unknown. Considering a statistical power of 80% and a hypothesized large effect size of 1 standard deviation, it was calculated that 48 participants (24 on VD and 24 on MD) would be needed to detect a statistical difference of *p* < 0.05 between the groups (nQuery, Statistical Solutions, Cork, Ireland). At least four additional participants were randomized as a reserve for drop-outs.

Values of biomarkers at baselines were checked for normal distribution by Kolmogorov-Smirnov-Test. Depending on distribution, differences between groups (MD versus VD) at the beginning were tested by unpaired *t*-test or Mann-Whitney-U-Test. Of the biomarkers investigated in this study, Met, MSO, and 25-OH-vitamin D_2_/D_3_ exhibited a normal distribution.

To investigate whether biomarkers (B_12_ metabolism, inflammation, and cardiovascular disease) changed from baseline to end of study upon dietary intervention, a mixed ANOVA was performed to test for the between-subject factor of *Diet* (two levels, meat-rich and plant-based), and the within-subject factor of *Time* (two levels, baseline and end). In case of significant interaction, a post hoc test (paired *t*-test if normally distributed, or Wilcoxon’s test if not normally distributed) with Bonferroni correction was conducted.

To compare the values of biomarkers at the end of dietary intervention, an ANCOVA was performed using the baseline values as covariates to ensure that differences at the end of the study are independent of the differences between the groups at baseline.

Correlation was calculated by Spearman-Rho. tHcy and Cysteine (Cys) were log_10_ transformed for the determination of tHcy:Cys ratio. Statistical analysis was performed using IBM SPSS (version 25.0). The significance level was set to α = 0.05.

## 3. Results

### 3.1. Characteristics of the Participants

In total, 150 interested persons were screened for eligibility by phone call. 103 persons were invited to a preparatory interview, and 61 persons were included in the study and started the run-in phase. From these, eight had to be excluded before randomization because of late withdrawal of consent or acute illness. Fifty-three participants completed the run-in phase, were randomized and started the intervention phase. Twenty-six participants were allocated to VD and 27 to MD for four weeks. All 53 completed the study as per protocol (Figure 2). Of the 53 participants who remained in their respective VD and MD regimens, nine of them (four VD and five MD) were excluded from the nutritional protocol assessment due to insufficient documentation of serving size and frequency. These participants remained eligible for biomarker analysis because none deviated from their assigned dietary intervention and all met criteria of inclusion concerning energy intake, body weight, and so forth. Demographics of participants are shown in Table 1.

### 3.2. Analysis of Vitamin B_12_ Status before and after 4-Week Intervention

Baseline serum vitamin B_12_ of VD group was on average 362.9 ± 110.9 ng/mL and decreased significantly to 296.1 ± 94.1 ng/mL in VD (*p* < 0.001), and the concentration of vitamin B_12_ of MD remained stable after the trial (*p* = 0.919; see also Table 2 and Figure 3, Panel C).

Baseline holo-TC of VD group was 67.3 ± 23.5 pmol/L and decreased significantly to 43.6 ± 20.0 pmol/L (*p* < 0.001). Twenty-five out of 26 vegans showed a decrease of holo-TC (−23.7 pmol/L). Baseline holo-TC of MD group was 69.7 ± 29.7 pmol/L and decreased significantly to 64.4 ± 28.7 pmol/L (*p* = 0.041; see also Table 2 and Figure 3, Panel A). In the MD group, 17 of 27 participants showed a decrease of holo-TC (−5.3 pmol/L).

A positive correlation was identified between serum holo-TC and vitamin B_12_ concentrations in both dietary groups examined after intervention (VD: *r* = 0.695, *p* < 0.001; MD: *r* = 0.484, *p* = 0.010), shown in Figure 4. Figure 4 depicts the linear fit of correlation analysis of VD (Panel A) and MD (Panel B) at the end of the trial.

Baseline and end values of MMA (Table 2 and Figure 3, Panel B) as well as of tHcy (Table 3 and Figure 5, Panel D) did not differ significantly between the groups.

Biomarkers of B-vitamin status and related metabolites (Cystathionine, Cysteine, Glutathione, Methionine, Methionine sulfoxide) did not differ significantly between the groups (Table 3 and Figure 5).

We next examined correlations between cellular biomarkers of vitamin B_12_, tHcy and MMA, and related B-vitamin metabolites with serum vitamin B_12_. While tHcy and MMA are measured in plasma, these metabolites are exclusively produced by cells. Therefore, the concentration of tHcy and MMA in plasma represents cellular production and export of these metabolites into circulation upon intracellular utilization of vitamin B_12_ by methionine synthase and methylmalonyl-CoA mutase. This is in contrast to serum vitamin B_12_ and holo-TC that derive from dietary intake, intestinal absorption and transport into circulation. The four biomarkers of vitamin B_12_ were examined for correlations. Negative correlations were found for serum vitamin B_12_ and methylmalonic acid (*r* = −0.353, *p* = 0.010; Figure 6, Panel B) and serum vitamin B_12_ and homocysteine (*r* = −0.280, *p* = 0.042; Figure 6, Panel A).

### 3.3. Proportion of Individuals with Abnormal Serum Parameters of Vitamin B_12_ Status after the 4-Week Trial

An overview of abnormal values (values outside the reference range) of vitamin B_12_ status (MMA, tHcy, holo-TC and serum vitamin B_12_) at the end of the trial is shown in Table 4. Values of 20 subjects (38%; 11 VD and 9 MD) were outside the reference range at the end of the trial. Vitamin B12 was lower than normal range (<200 ng/L) in six subjects (4 VD and 2 MD, *p* = 0.426), and holo-TC was lower than normal range (<35 pmol/L) in 10 subjects (7 VD and 3 MD, *p* = 0.185).

MMA was above normal range (>300 nmol/L) in seven subjects at the beginning of the trial, and in 10 subjects (5 VD and 5 MD, *p* = 0.500) at the end of the trial. Only three of the subjects with MMA above normal range at the trial beginning had also elevated levels at the end of the trial.

Before the dietary intervention, three subjects had hyperhomocysteinemia (>15 µmol/L), which remained elevated above normal range in two subjects (1 VD and 1 MD, *p* = 0.098) at the end of the trial.

None of the subjects showed clinical symptoms of vitamin B_12_ deficiency, neither at the beginning nor at the end of the trial.

### 3.4. Analysis of Homocysteine Metabolism before and after the 4-Week Trial

Dietary intake of folate and vitamin B_6_ were within the recommended range (folate: 200–400 µg/day [26], vitamin B_6_: ♀ > 1.4 mg/day, ♂ > 1.6 mg/day [27]) in both randomized groups (Table 5).

As mentioned before, serum concentrations of tHcy, cystathionine, Cys, GSH, Met, and MSO did not differ significantly between the groups (Table 3 and Figure 5). In all cases, the serum concentration of Met, Cys, cystathionine and GSH were within the normal reference ranges reported for healthy adults [14,15].

An association has been described between the fraction of reduced Cys and Hcy with respect to the total concentrations of these metabolites (tCys and tHcy) in humans [16]. Due to sample collection protocols, we could only assess total concentrations of these metabolites, i.e., tCys and tHcy. We examined associations between tCys and tHcy and if any, whether this was influenced by dietary intervention. Plots of log tCys versus log tHcy are shown in Figure 7, Panel A at baseline and Panel B after intervention. Analysis of log tCys versus log tHcy in the VD group showed no statistically significant associations neither at baseline nor at the end of the trial (VD_baseline_: *r* = 0.272, *p* = 0.179; VD_end_: *r* = 0.043, *p* = 0.834). Examination of tCys versus log tHcy in the MD group retrieved no associations at baseline (MD_baseline_: *r* = 0.108, *p* = 0.591), but a statistically significant association at the end of the trial (MD_end_: *r* = 0.503, *p* = 0.007). This suggests that a diet rich in meat influences the metabolic relationship of cysteine with its precursor metabolite homocysteine.

Recent work by Ulvik et al. demonstrated that the ratios of serum tHcy:tCys and tHcy:tCys:creatinine are superior in assessing overall B-vitamin status compared to tHcy alone [28]. Baseline ratio of VD group was 0.038 ± 0.01 and baseline ratio of MD group was 0.043 ± 0.01, and at the end of the trial the ratio was 0.045 ± 0.01 in VD group and 0.040 ± 0.01 in MD group (Figure 7, Panel C). The lack of statistical significance demonstrates no differences between VD and MD concerning B-vitamin status at the end of the trial.

### 3.5. Analysis of 25-OH-Vitamin D_2_/D_3_

Baseline value of 25-OH-vitamin D_2_/D_3_ MD group was 29.6 ± 13.9 ng/mL and baseline value of VD group was 32.9 ± 15.6 ng/mL (*p* = 0.439). After the trial, 25-OH-vitamin D_2_/D_3_ of MD was 31.4 ± 11.3 ng/mL and 25-OH-vitamin D_2_/D_3_ of VD was 34.2 ± 15.3 (*p* = 0.434). There was a time-dependent increase of 25-OH-vitamin D_2_/D_3_ (*p* = 0.039), but there was no difference between the groups (see Table 2 and Figure 3, Panel D). Twelve participants had a deficiency of 25-OH-vitamin D_2_/D_3_ (<20 ng/mL) at the beginning of the trial, and deficiency remained in all of these participants at the end of the trial.

### 3.6. Plasma Nitrite and Nitrate

There was no *Time* effect (*p* = 0.860) but a significant *Diet* effect (*p* = 0.009) and a significant Time × Diet interaction (*p* = 0.024) for the concentration of nitrite and nitrate. Post hoc analysis revealed that concentration of total content of nitrate and nitrite did not changed significantly over time for the VD group (baseline: 13.5 ± 6.9 µM, end: 16.1 ± 9.9 µM, *p* = 0.367), but decreased significantly for the MD group (baseline: 12.0 ± 6.8 µM, end: 8.8 ± 5.5 µM, *p* = 0.010). Nitrite and nitrate concentration differed significantly between the VD and MD groups at the end of the trial (*p* = 0.003).

### 3.7. Urine Analysis

There was a significant *Diet* effect (*p* = 0.010) but no *Time* effect (*p* = 0.407) nor a significant Time × Diet interaction (*p* = 0.949) for urine creatinine concentration. At baseline, the concentration of urine creatinine differed significantly between the VD group and the MD group (VD: 1.4 ± 0.9 mg/mL, MD: 1.9 ± 1.0 mg/mL, *p* = 0.029). After four weeks of intervention, the concentration decreased to 1.3 ± 0.7 mg/mL in the VD group and to 1.8 ± 1.1 mg/mL in the MD group. There were no significant differences in urine creatinine concentration between the VD group and the MD group at the end of the trial (*p* = 0.158).

There was no significant *Diet* effect (*p* = 0.986), nor *Time* effect (*p* = 0.208) nor a significant Time × Diet interaction (*p* = 0.310) for the PGEM values. There were no significant differences at baseline in between the VD group and the MD group (VD: 23.0 ± 43.4 ng/mL, MD: 31.5 ± 34.6 ng/mL, *p* = 0.083). After four weeks of intervention the concentration of PGEM was similar to baseline values (VD 23.1 ± 40.4 ng/mL and MD 31.0 ± 34.2 ng/mL). There were no significant differences in PGEM values between the VD group and the MD group at the end of the trial (*p* = 0.676).

### 3.8. Evaluation of Nutritional Protocols

From the 53 subjects, three did not fill in the standardized nutrition protocol, but used their own protocol, and another six subjects reported an intake of on average less than 1100 kcal daily, which was not credible (BMI stable). Therefore, these nine participants were excluded from analysis of nutritional protocols, but not from analysis of biomarkers (Figure 2). Overall, dietary intake of 44 (22 VD, 22 MD) was analyzed. We found no significant group difference of baseline values (Table 5). Daily intake of energy was similar in both groups (VD mean 2240.8 ± 894.8 kcal versus MD mean 2242.3 ± 762.7 kcal) during the trial. A detailed analysis of micronutrients showed significant differences between VD and MD. All results are shown in Table 5.

### 3.9. Nutritional Assessment before and after 4-Week Intervention

Dietary intake of nutrients and micronutrients. Cholesterol, protein, and fat intake in the VD group were within recommended ranges for healthy subjects. Protein and fat intake of the MD group were within recommended allowances; however, for a significant proportion of subjects (91%), the daily cholesterol intake (mg) exceeded the daily allowance of 300 mg recommended by the FDA (U.S. Food and Drug Administration, USA). The daily intake of cholesterol of subjects in the MD regime was 324.8 ± 140.3 mg before the trial and increased significantly to 452.6 ± 185.5 mg (*p* < 0.001). The daily intake of cholesterol of subjects in the VD regime was 332.9 ± 167.6 mg before the trial and decreased significantly to 28.7 ± 25.6 mg (*p* < 0.001). The daily intake of protein and fat of subjects in the MD group increased significantly during the trial (*p* = 0.001 and *p* = 0.012, respectively), whereas the daily intake of protein and fat of subjects in the VD group decreased significantly (*p* = 0.028 and *p* = 0.046, respectively). All results are shown in Table 5.

During the trial, the proportions of saturated and monounsaturated fatty acids in VD and MD was different: in the VD group, the intake of saturated fatty acids decreased significantly during the trial (*p* < 0.001), and the intake of MD group increased significantly (*p* = 0.003; Table 5). The intake of monounsaturated fatty acids of VD group remained stable during the trial (*p* = 0.099), whereas intake of MD group increased significantly (*p* = 0.007; Table 5).

The intake of fiber of MD group decreased significantly (*p* = 0.014), and the intake of the VD group increased significantly (*p* < 0.001). The intake of salt, copper, folate, and vitamin C of the MD group remained stable during the trial (*p* = 0.058, *p* = 0.664, *p* = 0.131, *p* = 0.181, respectively), whereas the VD intake of salt decreased significantly (*p* = 0.014) and the intake of copper, folate, and vitamin C increased significantly (*p* = 0.008, *p* = 0.013, *p* = 0.017, respectively). The intake of phosphorus and vitamin B_3_ increased significantly in the MD group (*p* = 0.05), but both remained stable in the VD group (*p* = 0.115 and *p* = 0.390, respectively). In the MD group, the intake of phenylalanine increased significantly during the trial (*p* < 0.001) and the intake of phenylalanine of the VD group decreased significantly (*p* = 0.019).

VD group showed as expected significantly lower intake of vitamin B_12_ (Table 5). The intake of vitamin B_12_ decreased significantly in VD group (*p* < 0.001) and increased significantly in MD group (*p* < 0.001). Nutritional protocols revealed further two critical micronutrients in the VD group as measured values were marginally lower than the recommendation of the German Nutrition Association (DGE) [20]: intake of zinc was on average 9.0 ± 5.3 mg per day (reference value > 10 mg/day) and the daily intake of vitamin B_2_ was 1.3 ± 0.7 mg (reference value > 1.4 mg/day). In the VD group, the baseline of daily intake of zinc was 10.8 ± 4.2 mg and decreased significantly to 9.0 ± 3.9 mg after the trial (*p* < 0.001). In the MD group, the baseline of daily intake of zinc was 11.2 ± 4.7 mg and increased significantly to 14.7 ± 6.7 after the trial (*p* < 0.001). The intake of B_2_ of VD group was 1.6 ± 0.6 mg and decreased significantly to 1.3 ± 0.7 mg after the trial (*p* = 0.017). The intake of B_2_ of MD group was 1.5 ± 0.6 mg and increased significantly to 2.0 ± 0.8 after the trial (*p* = 0.001).

Iodine intake was deficient (<200 µg) in both diets (VD 62.6 ± 23.3 versus MD 119.1 ± 52.8, see Table 5). The intake of iodine in VD group decreased significantly during the trial (*p* < 0.001), the intake of MD group remained stable (*p* = 0.543).

### 3.10. Fatty Acid Profiles

Results of fatty acids measurements in serum are shown in Table 6 and Figure 8. Both groups exhibited comparable profiles of selected serum fatty acids, with the only exception being arachidonic acid. Baseline value of arachidonic acid of VD group remained stable (*p* = 0.424). Baseline value of arachidonic acid of MD group was 3.7 ± 1.8 µM and increased significantly to 6.4 ± 8.0 µM (*p* = 0.021; see also Table 6 and Figure 8, Panel A).

## 4. Discussion

The number of individuals transitioning into plant-based nutrition is growing worldwide. This change in dietary habits may involve environmental, philosophical, and health reasons. Among the documented benefits of plant-based diets are early and recent studies demonstrating the prevention and reversal of cardiovascular disease [29,30,31] and the significant alleviation of symptoms in inflammatory diseases such as rheumatic disorders [32,33]. One major concern of plant-based diets is the occurrence of vitamin B_12_ deficiency, as this micronutrient is not found in plants, and not all plant-based individuals supplement their diets with oral vitamin B_12_. While it takes approximately 2 to 5 years to exhaust body stores of vitamin B_12_, and therefore, to develop clinical symptoms of vitamin B_12_ deficiency [16], very little is known about the effects of short-term reduction of vitamin B_12_ intake. This study aimed to answer two major scientific questions: (a) Do systemic markers of vitamin B_12_ status respond to short-term dietary intervention with a meat-rich or a plant-based diet? and (b) Do meat-rich and vegan diets have an impact on plasma markers of inflammation and cardiovascular disease?

### 4.1. Effect of Diet on Markers of Vitamin B_12_ Status

There are four canonical markers of vitamin B_12_ status: Hcy, MMA, holo-TC, and vitamin B_12_. These biomarkers are measured routinely in plasma and serum and examined with respect to reference ranges [6]. Hcy and MMA are exclusively produced by cells upon metabolization of vitamin B_12_. Therefore, all Hcy and MMA that build up in plasma derive from cellular metabolism. This is different from biomarkers holo-TC and serum vitamin B_12_, which derive from dietary intake, intestinal absorption, and transport into circulation, not requiring enzymatic utilization in the cell. Hcy and MMA become elevated when the bioactive fraction of circulating vitamin B_12_, holo-TC, is low enough to compromise the activities of cellular methionine synthase and methylmalonyl-CoA mutase. Elevation of Hcy and MMA indicate clinical deficiency of vitamin B_12_. Because reduction of holo-TC precedes elevation of Hcy and MMA, holo-TC constitutes an interesting biomarker to monitor vitamin B_12_ status and adequacy of intake before the onset of clinical vitamin B_12_ deficiency.

In 2013, Carmel raised the question “If an individual has temporary disruption (e.g., for a week or two) of vitamin B_12_ assimilation, either because intake is transiently reduced or because one of the many drugs that transiently interfere with vitamin B_12_ absorption is used, does plasma holotranscobalamin fall even though such brief disruptions cannot affect vitamin B_12_ stores or status significantly?”, which has not been answered to date [16]. In this study, we showed that short-term reduced intake of vitamin B_12_ does lead to a rapid (i.e., 4 weeks) decrease of serum holo-TC. These findings provide an answer to the aforementioned diagnostic question [16], and demonstrate that short-term disruption of vitamin B_12_ intake leads to a rapid decrease of holo-TC concentration without effect on cellular markers of vitamin B_12_ status Hcy and MMA. The correct assessment of vitamin B_12_ status has been a matter of intense discussion. The status of vitamin B_12_ is not static; it varies over time [16] as well as by gender and age [34]. In addition, serum vitamin B_12_ only reflects circulating vitamin B_12_ and does not necessarily represent the cellular status of vitamin B_12_ [6,35]. Holo-TC as a transport protein is subject to genetic polymorphisms [36,37,38] and also depends on kidney function [37]. Elevation of the functional indicators MMA and tHcy indicates lack of vitamin B_12_ within the cells [34]. MMA is also elevated in patients with chronic kidney disease [37,39]. Our cohort of healthy individuals was free from this confounder factor that influences MMA concentration in serum. The sole elevation of MMA (>300 nmol/L) of five participants (Table 3) at the end of the trial might be attributable to an already described temporal fluctuation of MMA, as it occurred independently of the assigned diet [40]. Two participants (1 VD, 1 MD) showed elevated tHcy (>15 µmol/L) at the beginning and at the end of the trial unrelated to the assigned diet (Table 3). Deficiency of folate and vitamin B_6_ lead to an increase of tHcy [37], but both VD and MD groups in our study exhibited adequate intake of folate and vitamin B_6_ and correspondingly normal metabolite profiles.

### 4.2. Value of Holo-TC for the Assessment of Vitamin B_12_ Status with Respect to Diet

While onset of vitamin B_12_ deficiency in vegan subjects is a point of serious concern, the window of time for its prevention is rather safe. Depending on liver function, pre-existing diseases, previous amount of vitamin B_12_ intake, and vitamin B_12_ turnover, it takes approximately 2–5 years to exhaust liver and kidney reserves of vitamin B_12_ to reach a status of clinical vitamin B_12_ deficiency [16,41,42]. It is therefore reasonable to pose the question: is holo-TC, the biologically active form of vitamin B_12_, a reliable marker of vitamin B_12_ status in individuals with a vitamin B_12_-deficient diet? Our results are in line with previous reports showing that holo-TC is lower in vegans [39,43] and that this biomarker decreases earlier in time than cellular markers of vitamin B_12_ deficiency, Hcy, and MMA [39]. This supports the proposal that holo-TC may be a reliable marker to monitor vitamin B_12_ intake and status in individuals at risk, including vegans, before the development of clinical vitamin B_12_ deficiency. The major sources of TC biosynthesis are thought to include liver, intestinal cells, and the vascular endothelium [19,43,44,45,46,47], but little is known about the regulation of TC biosynthesis with respect to intake of vitamin B_12_ [16]. Quadros et al. reported that synthesis of holo-TC in enterocytes of guinea pigs is mediated by the complex of vitamin B_12_ and its transport protein intrinsic factor [47]. It is known that holo-TC rapidly increases after oral, intravenous, or intramuscular administration of vitamin B_12_ in humans [11,48,49,50]. Holo-TC might be an indicator for vitamin B_12_ absorption [12]. Because holo-TC is a specialized transporter of vitamin B12, it is possible that its concentration in plasma is dictated by nutritional intake of the micronutrient [16]. In our study, 85% of the subjects maintained holo-TC within the normal reference range (>35 pmol/L [34]) and metabolic markers did not show deficiency of vitamin B_12_. Our results are in accord with a study performed in a cohort of life-long unsupplemented healthy vegetarian Indian graduates (♀ 61%, on average 29 years old, *n* = 119) where holo-TC was shown to be a good biomarker for monitoring dietary intake of vitamin B_12_, but not of clinical evident vitamin B_12_ deficiency as none of the participants showed signs of deficiency [15].

### 4.3. Biomarker Associations and Cut-Off Values for the Assessment of Vitamin B_12_ Status in Plant-Based Subjects

An inverse correlation was identified between MMA and vitamin B_12_. The joint examination of biomarkers of vitamins B_6_, B_9_, and B_12_ pathways demonstrated that overall Hcy metabolism was unaffected by 4-weeks of dietary intervention with either vegan or meat-based diets.

However, our observations are limited by the short-term nature of our trial and may not be valid to describe vitamin B_12_ status in long-term vegetarians and vegans. For example, examination of vitamin B_12_ status in the aforementioned cohort of 119 young, healthy, unsupplemented vegetarian Indian graduates showed that 50% were vitamin B_12_ deficient (<148 pmol/L), 50–70% exhibited low plasma holo-TC (<34 pmol/L) and 70–90% presented elevated plasma total Hcy (>15 μmol/L) [15]. The lack of clinical signs of B_12_ deficiency and results from ROC curves led the authors of the study to propose the use of the following adjusted plasma cut-off values for subjects adhering to life-long vegetarianism in India: vitamin B_12_: 105 pmol/L, holo-TC: 22.6 pmol/L and tHcy 17.6 μmol/L for females and 27.0 μmol/L for males [15]. The study posited that applying reference values and cut-offs established from studying omnivorous populations, i.e., subjects with a daily intake of vitamin B_12_ that keeps tHcy and MMA to minimum levels and holo-TC above 35 pmol/L, may lead to the over-diagnosis of vitamin B_12_ deficiency in vegetarians and vegans who experience a significantly lower dietary intake of vitamin B_12_ without it necessarily representing a health threat [15,51].

Our short-term intervention demonstrated rapid reduction of holo-TC upon lower intake of vitamin B_12_ in the vegan diet group. This did not manifest in overt signs of vitamin B_12_ deficiency such as anemia or neurological/neuropsychiatric symptoms [6]. Long-term follow up is necessary to determine whether this reduction in holo-TC continues to fall or if it reaches a plateau despite a sustained low intake of the micronutrient. This would help to address whether adjusted reference ranges for biomarkers should indeed be employed routinely when examining vitamin B_12_ status in plant-based individuals [51].

### 4.4. Effect of Diet on Metabolic Markers of Inflammation and Cardiovascular Health

Among the benefits of plant-based nutrition are associations with a lower risk for chronic inflammatory diseases. For example, plant-based nutrition reduced cardiovascular mortality compared to a mixed diet [29] and showed beneficial effects on the inflammatory activity in patients with rheumatic diseases [31,33,52]. It is assumed that a plant-based diet is free of arachidonic acid, an omega 6-fatty acid, which in humans and animals can be synthesized via linoleic and linolenic acid [53,54]. Arachidonic acid is, amongst others, a substrate for cyclooxygenases, which undergoes metabolization to form pro-inflammatory prostanoids such as prostaglandin E2 [53,55,56]. Volunteers of the VD group had significantly lower levels of arachidonic acid compared to the MD group. Arachidonic acid can be metabolized to pro-inflammatory messengers playing an important role in atherosclerosis and auto-immunity [33,56]. It has been reported that diets rich in arachidonic acid lead to an increase of pro-inflammatory, pro-thrombotic, and pro-atherogenic messengers [57,58]. Reduced intake of arachidonic acid might, therefore, at least in part explain the clinically observed decrease of inflammation after starting a vegan diet in patients with rheumatic diseases [31,33]. Although arachidonic acid can be synthesized from linoleic and linolenic acid, an increased intake of those precursors does not seem to increase arachidonic acid synthesis [58]. The regulation of endogenous synthesis of arachidonic acid remains a matter of study. Our nutritional assessment suggests that plant-based nutrition provides adequate precursor linoleic and linolenic acids to sustain homeostasis of this metabolic pathway [58,59].

Aside from fatty acid metabolism, a higher intake of nitrate in plant-based diets might also be favorable with regard to cardiovascular risk factors [60,61]. Results from our study showed that nitrite and nitrate concentration differed significantly between the VD and MD groups at the end of the trial (*p* = 0.003). Nitrate has a beneficial impact on blood pressure and vascular function [60,62]. Our findings thus point to beneficial effects of a vegan diet in supporting cardiovascular health [29,30].

### 4.5. Strengths and Limitations of This Study

For feasibility reasons, the sample size of our study was relatively small. Our study did not include a supplementation phase, whereby the response of holo-TC to controlled, increased intake of vitamin B_12_ could be assessed. A new clinical trial is in preparation to address this important aspect. The entire study was performed according to CONSORT (Consolidated Standards of Reporting Trials) and the Guideline for Good Clinical Practice. All subjects were randomized immediately before start of trial and were treated the same way (same time for information, nutritional training as well as sample collection, same number of clinician-participant contacts during the trial) except of nutritional allocation. The kind of food (vegan or meat-rich) recorded in the nutritional protocols (also in the nine excluded individuals) was suitable to the assigned diet indicating compliance of the participants. However, the evaluation of daily intake has intrinsic limitations because the nutritional protocol records “servings”, which can be easily mismeasured by participants. We were able to ensure that body weight remained stable among participants. Results were handled and analyzed blinded. The clinical usefulness and reliability of holo-TC as a fast-responding marker of dietary intake of vitamin B_12_ awaits studies in larger populations that include plant-based individuals.

## 5. Conclusions

Results from nutritional assessment, blood, and urine parameters suggest that plant-based nutrition is advantageous to health by featuring significant reduction in the dietary intake of cholesterol and adequate profiles of nutrient and micronutrient status. A major health concern of individuals adhering to a vegan diet without supplementation is the risk for vitamin B_12_ deficiency. Our data show that holo-TC, the bioactive form of vitamin B_12_ in plasma, decreased in only 4 weeks of low intake of vitamin B_12_. During this time, Hcy and MMA remained within normal reference ranges, and there were no clinical signs of vitamin B_12_ deficiency. Thus, plasma holo-TC may be a fast-responding biomarker to monitor adequate supply of vitamin B_12_ in plant-based individuals. We would like to bring attention to the difference between low holo-TC as an indicator of low intake of vitamin B_12_, versus holo-TC as a biomarker of vitamin B_12_ deficiency. A reduced intake of vitamin B_12_ leads to lower concentration of holo-TC in serum, which signifies lower bioactive vitamin B_12_ available to supply cells. This does not equate to cellular vitamin B_12_ deficiency, which is accompanied by elevation of biomarkers Hcy and MMA. The diagnosis of vitamin B_12_ deficiency should always include as a minimum one systemic biomarker (serum vitamin B_12_ or holo-TC) and one cellular biomarker (Hcy or MMA, preferable MMA). Therefore, holo-TC may be sufficient as a standalone biomarker to monitor intake of vitamin B_12_ in plant-based individuals. In contrast, if suspicion of vitamin B_12_ deficiency exists, we recommend the concerted measurement of holo-TC with at least one biomarker of cellular B_12_ status, preferably MMA.

## Figures and Tables

**Figure 1 nutrients-11-02815-f001:**
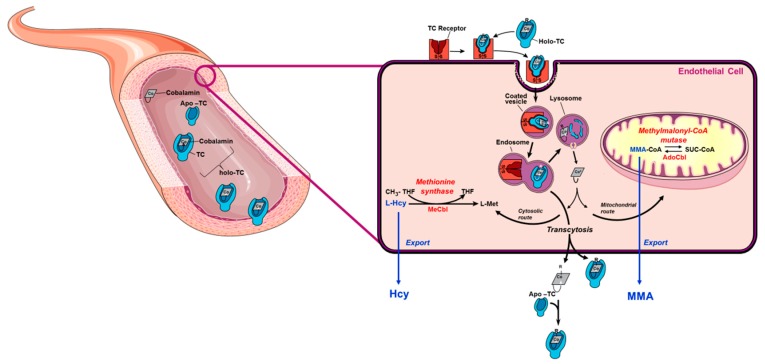
Vitamin B_12_ metabolism. Dietary vitamin B_12_ is transported through the digestive system and absorbed in the intestine where it binds to apo-transcobalamin (apo-TC) to form holotranscobalamin (holo-TC). Holo-TC is the bioactive form of vitamin B_12_. Holo-TC reaches circulation and it is then taken up by all cells in the body via receptor-mediated endocytosis (the transcobalamin (TC) receptor is also known as CD320). Once in the cell, vitamin B_12_ is freed from TC in the lysosome and exported into the cytosol. Downstream cytosolic processing and trafficking events ensure that vitamin B_12_ ultimately reaches cytosolic methionine synthase and mitochondrial methylmalonyl-CoA mutase. Nutritional deficiency of vitamin B_12_ blocks the reactions catalyzed by methionine synthase and methylmalonyl-CoA, resulting in the accumulation of their respective substrates, homocysteine (Hcy) and methylmalonyl-CoA (MMA-CoA). Hcy and MMA are toxic to cell metabolism; therefore, cells export these metabolites into circulation under conditions of B_12_ deficiency. This manifests clinically as elevated concentration of Hcy and MMA in plasma or serum. A vascular endothelial cell is used as a generic model. This figure was modified from reference [19].

**Figure 2 nutrients-11-02815-f002:**
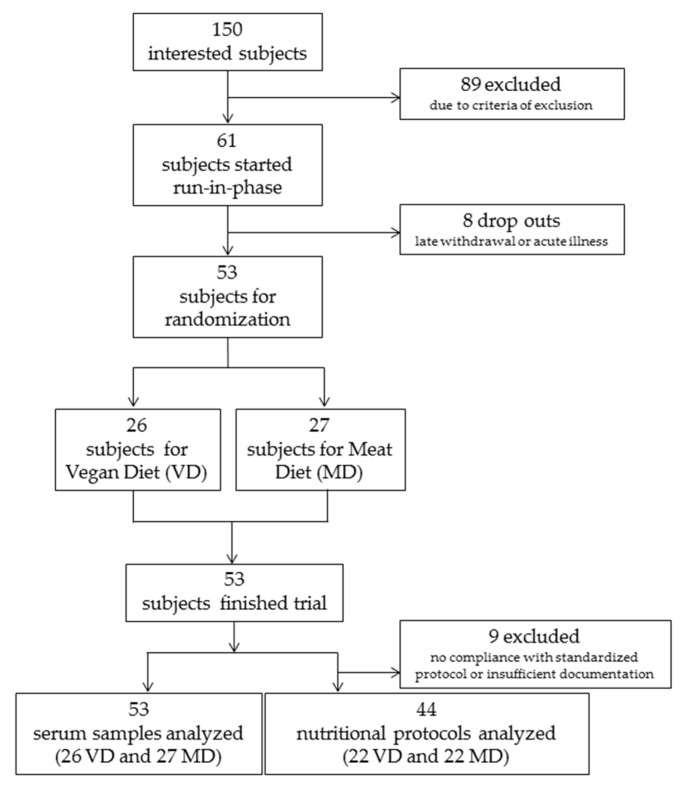
Flow chart depicting randomization scheme of participants into vegan (VD) and meat-rich diets (MD).

**Figure 3 nutrients-11-02815-f003:**
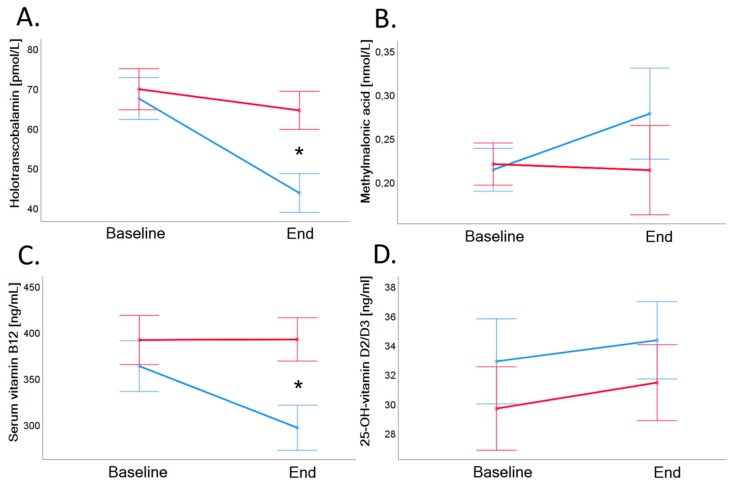
Holotranscobalamin (Panel **A**), methylmalonic acid (Panel **B**), vitamin B_12_ (Panel **C**), and 25-OH-Vitamin D_2_/D_3_ (Panel **D**) in serum before and after 4-week dietary intervention in VD (blue line) and MD (red line). Serum vitamin B_12_ (ng/L) and holotranscobalamin concentrations (pmol/L) showed a significantly decrease in vegans (vitamin B_12_: 362.8 ± 110.9 to 296.1 ± 94.1 ng/L, *p* < 0.001; holotranscobalamin: 67.3 ± 23.5 to 43.6 ± 20.0 pmol/L, *p* < 0.001) compared to meat-rich subjects (vitamin B_12_: 391.2 ± 159.2 to 391.8 ± 143.0 ng/L, *p* = 0.919; holotranscobalamin: 69.7 ± 29.7 to 64.4 ± 28.7 pmol/L, *p* = 0.041). Holotranscobalamin decreased by more than 30% upon dietary intervention, whereas serum vitamin B_12_ decreased by 18%_._ Error bars show ± 1 standard error. Baseline and end values as well as statistical comparisons are shown in Table 2. Significant differences between groups are marked with an asterisk (*: *p* < 0.05).

**Figure 4 nutrients-11-02815-f004:**
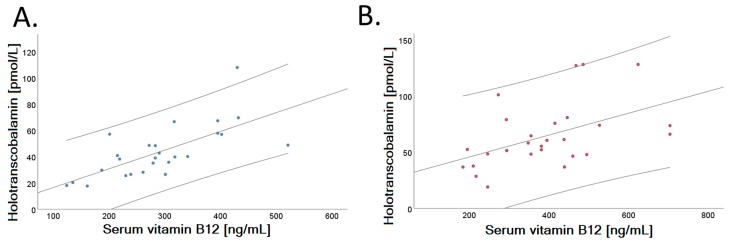
Relationship of serum holotranscobalamin and vitamin B_12_ in the two randomized groups after a 4-week dietary intervention. Vitamin B_12_ (ng/L) correlates positively with holotranscobalamin concentration (pmol/L) both after intervention in VD (Panel **A**, *r* = 0.695, *p* < 0.001) and in MD (Panel **B**, r = 0.484, *p* = 0.010). Dispersion lines show the 95% confidence interval of the linear regression fit.

**Figure 5 nutrients-11-02815-f005:**
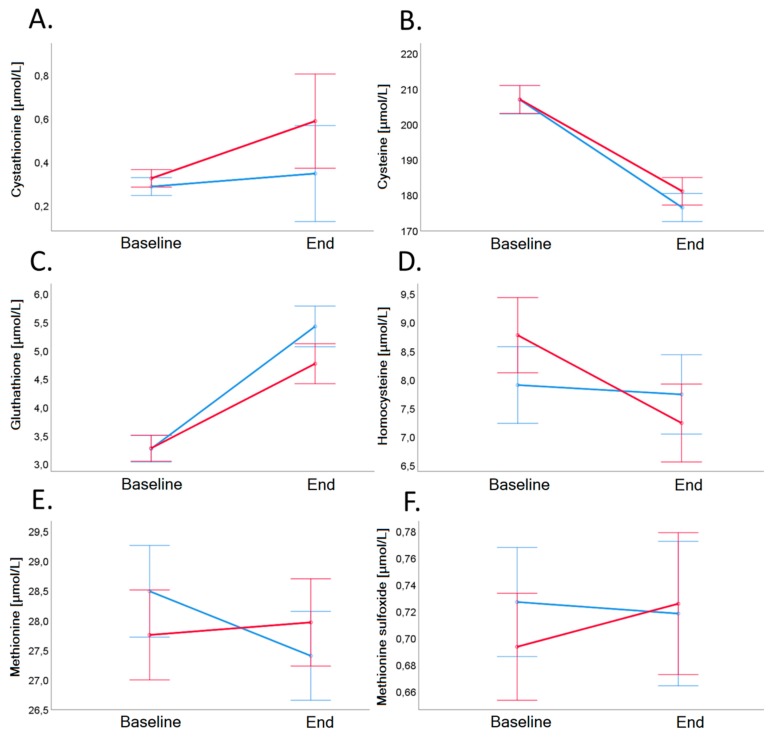
Cystathionine (Panel **A**), cysteine (Panel **B**), glutathione (Panel **C**), homocysteine (Panel **D**), methionine (Panel **E**), and methionine sulfoxide (Panel **F**) in plasma before and after 4-week dietary intervention in VD (blue line) and MD (red line). Error bars show ± 1 standard error. Baseline and end values as well as statistical comparisons are shown in Table 3.

**Figure 6 nutrients-11-02815-f006:**
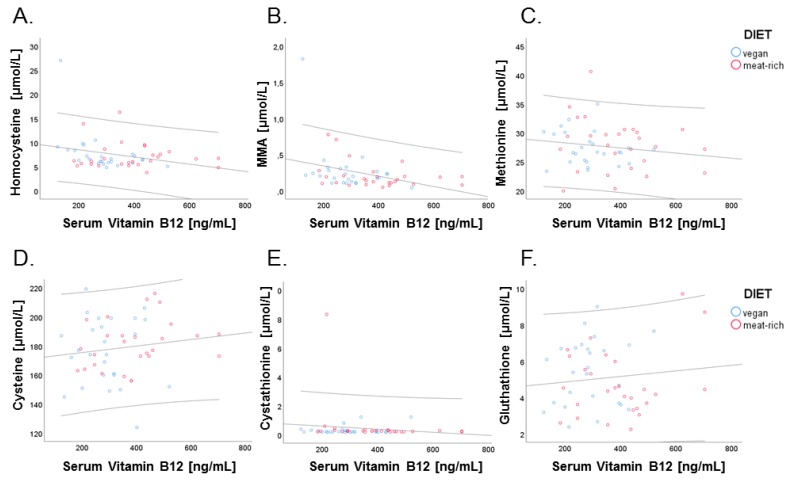
Relationship of serum vitamin B_12_ with cellular markers of vitamin B_12_, tHcy, and MMA, and with other marker metabolites of B-vitamin status. Data points of participants in the vegan and meat-rich diets are shown as blue and red dots, respectively. Correlations are shown for serum vitamin B_12_ (ng/L) and total homocysteine (µmol/L; *r* = 0.280, *p* = 0.042) (Panel **A**), serum vitamin B_12_ and methylmalonic acid (µmol/L; *r* = 0.280, *p* = 0.042) (Panel **B**), serum vitamin B_12_ and methionine (µmol/L; *r* = −0.154, *p* = 0.272) (Panel **C**), serum vitamin B_12_ and serum cysteine (µmol/L; *r* = 0.147, *p* = 0.293) (Panel **D**), serum vitamin B_12_ and cystathionine (µmol/L; *r* = −0.126, *p* = 0.367) (Panel **E**), and serum vitamin B_12_ and glutathione (µmol/L; *r* = 0.110, *p* = 0.434) (Panel **F**). Dispersion lines show the 95% confidence interval of the linear regression fit.

**Figure 7 nutrients-11-02815-f007:**
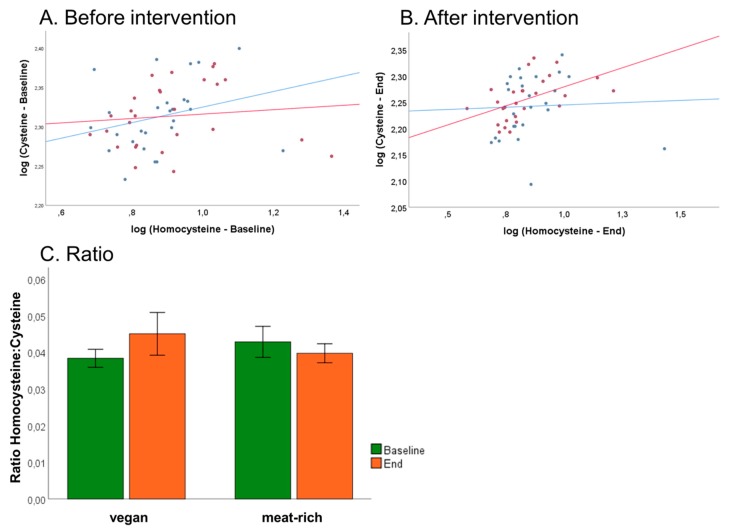
Associations of total cysteine and total homocysteine and influence of diet on B-vitamin status of VD (blue) and MD (red). (Panel **A**) Correlation of log of total cysteine and log of total homocysteine before trial. (Panel **B**) Correlation of log of total cysteine and log of total homocysteine after trial. (Panel **C**) Assessment of B-vitamin status using the tHcy:tCys ratio proposed by Ulvik et al. [17] at baseline and after 4-week intervention, sorted by assigned diet. Ratio shows non-significantly decrease in vegan group and non-significantly increase in meat-rich group, sorted by time (green baseline, orange end); error bars show ± 1 standard error.

**Figure 8 nutrients-11-02815-f008:**
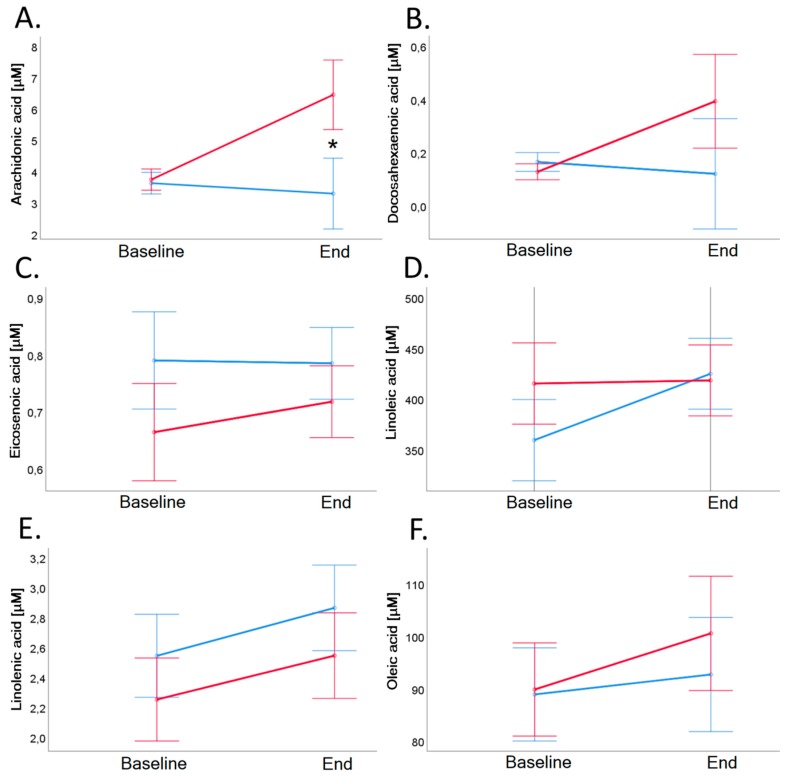
Arachidonic acid (Panel **A**), docosahexaenoic acid (Panel **B**), eicosenoic acid (Panel **C**), linoleic acid (Panel **D**), linolenic acid (Panel **E**), and oleic acid (Panel **F**) in serum before and after 4-week dietary intervention in VD (blue line) and MD (red line). Baseline value of arachidonic acid of MD group was 3.7 ± 1.8 µM and increased significantly to 6.4 ± 8.0 µM (*p* = 0.021), arachidonic acid of VD group remained stable (*p* = 0.424). Error bars show ± 1 standard error. Baseline and end values as well as statistical comparisons are shown in Table 3. Significant differences between groups are marked with an asterisk (*: *p* < 0.05).

**Table 1 nutrients-11-02815-t001:** Demographic data of participants in vegan diet group and in meat-rich group (SD = Standard deviation, *p*-value from *t*-test).

	Vegan ± SD (n = 26)	Meat-Rich ± SD (*n* = 27)	*p*-Value
Age (years)	33.2 ± 11.2	29.9 ± 9.5	0.407
Baseline: Body mass index (kg/m^2^)	22.9 ± 2.2	23.3 ± 2.6	0.444
End of study: Body mass index (kg/m^2^)	22.7 ± 2.0	23.4 ± 2.6	0.240
Gender (male/female, %)	31/69	44/56	0.309

**Table 2 nutrients-11-02815-t002:** Results of serum levels of holotranscobalamin, methylmalonic acid, vitamin B_12_, and vitamin D (25-OH-vitamin D_2_/D_3_) before and after intervention. SD = Standard deviation; *p*-value (baselines) indicates comparison of values at baseline (results from *t*-test or MWU-test). *p*-value (end) indicates comparisons at the end of the study adjusted for baseline values (results from ANCOVA). *p*-value (baseline-to-end) indicates comparisons of changes from baseline to end of study (results from mixed ANOVA). Statistically significant results are bolded.

	Baseline			End			*p*-Value (Baseline-to-End)		
	Vegan ± SD (*n* = 26)	Meat-Rich ± SD (*n* = 27)	*p*-Value (Baselines)	Vegan ± SD (*n* = 26)	Meat-Rich ± SD (*n* = 27)	*p*-Value (End)	Time	Diet	Time × Diet
Holotranscobalamin (pmol/L)	67.3 ± 23.5	69.7 ± 29.7	0.901	43.6 ± 20.0	64.4 ± 28.7	**<0.001**	**<0.001**	0.098	**<0.001**
Methylmalonic acid (nmol/L)	214.5 ± 129.6	220.0 ± 121.0	0.755	277.8 ± 330.0	213.1 ± 182.2	0.090	0.253	0.573	0.158
Serum vitamin B_12_ (ng/mL)	362.8 ± 110.9	391.2 ± 159.2	0.510	296.1 ± 94.1	391.8 ± 143.0	**<0.001**	**<0.001**	0.085	**<0.001**
25-OH-vitamin D_2_/D_3_ (ng/mL)	32.9 ± 15.6	29.6 ± 13.9	0.439	34.2 ± 15.3	31.4 ± 11.3	0.903	**0.039**	0.427	0.829

**Table 3 nutrients-11-02815-t003:** Amino acids associated with homocysteine metabolism measured in serum of the subjects in the vegan and meat-rich diet group after four weeks of intervention. SD = Standard deviation; *p*-value (baselines) indicates comparison of values at baseline (results from *t*-test or MWU-test). *p*-value (end) indicates comparisons at the end of the study adjusted for baseline values (results from ANCOVA). *p*-value (baseline-to-end) indicates comparisons of changes from baseline to end of study (results from mixed ANOVA). Statistically significant results are bolded.

	Baseline			End			*p*-Value (Baseline-to-End)		
	Vegan ± SD (*n* = 26)	Meat-Rich ± SD (*n* = 27)	*p*-Value (Baselines)	Vegan ± SD (*n* = 26)	Meat-Rich ± SD (*n* = 27)	*p*-Value (End)	Time	Diet	Time × Diet
Cystathionine (µmol/L)	0.3 ± 0.1	0.3 ± 0.3	0.378	0.3 ± 0.3	0.6 ± 1.5	0.642	0.213	0.446	0.430
Cysteine (µmol/L)	206.7 ± 21.3	206.8 ± 19.6	0.475	167.9 ± 29.7	183.3 ± 19.2	0.321	**<0.001**	0.641	0.378
Glutathione (µmol/L)	3.3 ± 1.1	3.3 ± 1.3	0.943	5.4 ± 1.8	4.8 ± 1.8	0.093	**<0.001**	0.395	0.090
Homocysteine (µmol/L)	7.9 ± 2.5	8.8 ± 4.1	0.606	7.7 ± 4.2	7.2 ± 2.7	0.140	**0.034**	0.832	0.084
Methionine (µmol/L)	28.5 ± 4.1	27.7 ± 3.8	0.631	27.2 ± 3.1	27.7 ± 3.9	0.415	0.472	0.922	0.288
Methionine sulfoxide (µmol/L)	0.7 ± 0.2	0.7 ± 0.2	0.466	0.7 ± 0.2	0.7 ± 0.3	0.707	0.748	0.818	0.578

**Table 4 nutrients-11-02815-t004:** List of subjects with abnormal serum parameters of vitamin B_12_ metabolism at the end of the trial and comparison to baseline values (*n* = 19). (+ shows abnormal value; holo-TC = holotranscobalamin, tHcy = Homocysteine, MD = meat-rich diet, MMA = Methylmalonic acid, VD = vegan diet).

Subject	Diet	B_12_ <200 ng/L	Holo-TC <35 pmol/L	MMA >300 nmol/L	tHcy >15 µmol/L	Abnormal Baseline Values
5	MD		+	+		
8	MD			+		
15	MD			+		
20	MD	+				
35	MD		+	+		MMA + tHcy +
39	MD				+	tHcy +
42	MD	+				B_12_ +
44	MD		+	+		MMA +
12	VD		+			
23	VD	+	+		+	tHcy +
24	VD			+		
25	VD		+			
27	VD		+			
29	VD			+		
34	VD	+	+			
41	VD			+		
48	VD	+	+	+		B_12_ + MMA +
49	VD	+	+			MMA +
51	VD			+		

**Table 5 nutrients-11-02815-t005:** Results of the nutritional protocols: comparison of mean of daily nutritional intake during four weeks of intervention. SD = Standard deviation; ^+^ due to food with added vitamin B_12_; *p*-value (baselines) indicates comparison of values at baseline (results from *t*-test or MWU-test). *p*-value (end) indicates comparisons at the end of the study adjusted for baseline values (results from ANCOVA). *p*-value (baseline-to-end) indicates comparisons of changes from baseline to end of study (results from mixed ANOVA). Statistically significant results are bolded (*p* < 0.05).

	Baseline			End			*p*-Value (Baseline-to-End)		
	Vegan ± SD (*n* = 22)	Meat-Rich ± SD (*n* = 22)	*p*-Value (Baselines)	Vegan ± SD (*n* = 22)	Meat-Rich ± SD (*n* = 22)	*p*-Value (End)	Time	Diet	Time × Diet
Intake of energy (kcal)	2228.8 ± 915.0	2333.7 ± 859.1	0.549	2240.8 ± 894.8	2242.3 ± 762.7	0.213	0.588	0.468	0.291
Kilojoule (kj)	9388.6 ± 3747.8	9771.1 ± 3593.1	0.565	9068.5 ± 2949.8	10265.6 ± 4210.7	0.211	0.586	0.470	0.287
Carbohydrates (g)	245.5 ± 95.4	256.9 ± 136.9	0.897	276.0 ± 85.1	241.5 ± 91.8	**0.001**	0.688	0.670	0.209
Percentage of energy intake	47%	47%		54%	41%				
Fat (g)	86.4 ± 42.2	89.3 ± 43.5	0.751	68.8 ± 29.1	106.3 ± 47.4	0.143	0.611	0.091	**0.002**
Percentage of energy intake	37%	36%		31%	40%				
Protein (g)	86.9 ± 33.4	94.0 ± 41.3	0.474	79.5 ± 28.5	112.4 ± 44.4	**<0.001**	0.413	**0.020**	**<0.001**
Percentage of energy intake	16%	17%		15%	19%				
Cholesterol (mg)	332.9 ± 167.6	324.8 ± 140.3	0.787	28.7 ± 25.6	452.6 ± 185.5	**<0.001**	**0.003**	**<0.001**	**<0.001**
Saturated fatty acids (g)	32.2 ± 15.0	34.1 ± 15.4	0.681	15.4 ± 5.7	43.9 ± 20.5	**<0.001**	0.309	**<0.001**	**<0.001**
Monounsaturated fatty acids (g)	31.6 ± 17.5	31.6 ± 17.5	0.769	26.9 ± 14.1	38.4 ± 17.3	**0.005**	0.370	0.345	**0.003**
Polyunsaturated fatty acids (g)	15.1 ± 8.8	15.2 ± 8.3	0.565	19.6 ± 7.9	15.0 ± 7.0	0.063	**0.025**	0.121	0.193
Fiber (g)	27.8 ± 10.8	34.5 ± 25.6	0.342	45.7 ± 19.5	24.9 ± 11.05	**<0.001**	0.171	0.077	**<0.001**
Salt (g)	4.6 ± 2.2	5.0 ± 1.8	0.255	3.8 ± 1.8	6.1 ± 3.2	**0.003**	0.589	**0.040**	**0.006**
Sugar (g)	118.5 ± 61.4	114.0 ± 70.7	0.664	114.4 ± 32.8	108.1 ± 52.46	0.585	0.527	0.669	0.855
Uric acid (mg)	488.4 ± 236.2	491.9 ± 194.6	0.842	603.0 ± 279.5	603.6 ± 276.1	0.849	**0.003**	0.840	0.754
Sodium (mg)	2196.0 ± 1479.4	2216.9 ± 803.7	0.405	1692.7 ± 802.9	2736.3 ± 1427.6	**0.002**	0.844	0.105	**0.007**
Potassium (mg)	3663.7 ± 1316.1	3527.0 ± 1039.8	0.474	4186.2 ± 1794.4	3605.6± 1671.0	0.401	0.160	0.209	0.610
Calcium (mg)	841.1 ± 378.6	854.6 ± 244.9	0.534	645.1 ± 390.1	933.8 ± 447.3	**0.029**	0.556	0.107	0.051
Magnesium (mg)	401.6 ± 181.3	397.5 ± 149.5	0.681	505.4 ± 213.2	388.2 ± 172.9	0.107	0.080	0.113	0.214
Copper (μg)	1935.6 ± 877.7	1838.8 ± 693.9	0.787	2564.0 ± 974.2	1763.9 ± 703.5	**0.008**	**0.018**	**0.011**	**0.040**
Fluoride (µg)	1079.4 ± 908.3	1027.5 ± 342.5	0.265	912.7 ± 358.5	1055.7 ± 477.3	0.249	0.663	0.840	0.314
Iodione (µg)	118.3 ± 41.3	116.5 ± 40.7	0.787	62.6 ± 23.2	119.1 ± 52.8	**<0.001**	**0.004**	**0.007**	**<0.001**
Iron (mg)	11.9 ± 4.5	11.9 ± 4.6	0.824	14.8 ± 6.1	13.5 ± 5.4	0.579	**0.004**	0.398	0.736
Phosphorus (mg)	1344.2 ± 454.1	1423.2 ± 517.1	0.405	1193.2 ± 494.2	1602.6 ± 677.6	**0.018**	0.510	0.120	**0.021**
Zinc (mg)	10.8 ± 4.2	11.2 ± 4.7	0.647	9.0 ± 3.9	14.7 ± 6.4	**<0.001**	0.109	**0.026**	**<0.001**
Retinol (mg)	1963.7 ± 1256.0	2396.6 ± 2423.4	0.879	1924.5 ± 1210.0	2464.9 ± 2024.3	0.323	0.640	0.406	0.549
β-carotene (µg)	9317.3 ± 4797.7	7666.9 ± 4749.1	0.405	1130.3 ± 723.3	681.4 ± 621.7	0.050	0.571	0.064	0.107
Vitamin B_1_ (mg)	1.6 ± 1.0	1.5 ± 0.6	0.725	2.0 ± 0.8	1.8 ± 0.9	0.687	**0.001**	0.228	0.860
Vitamin B_2_ (mg)	1.6 ± 0.6	1.4 ± 0.7	0.787	1.3 ± 0.7	2.0 ± 0.8	**0.003**	0.510	**0.040**	**0.004**
Niacin (mg)	3686.4 ± 1858.5	3724.7 ± 1789.8	0.842	3431.2 ± 1629.0	4588.1 ± 1933.5	**0.007**	0.082	0.345	**0.005**
Vitamin B_6_ (mg)	2.0 ± 0.9	2.0 ± 0.7	0.916	2.3 ± 1.1	2.2 ± 1.0	0.930	0.055	0.603	0.934
Folate (μg)	345.9 ± 187.1	327.3 ± 126.0	0.842	509.1 ± 317.0	286.7 ± 117.2	**0.007**	**0.038**	**0.012**	**0.010**
Vitamin B_12_ (μg)	4.8 ± 2.1	5.9 ± 5.4	0.769	0.8 ^+^ ± 1.2	8.5 ± 4.1	**<0.001**	0.602	**<0.001**	**<0.001**
Vitamin C (mg)	132.5 ± 77.1	131.8 ± 49.8	0.597	182.1 ± 78.9	119.5 ± 57.0	**0.006**	0.146	**0.032**	**0.037**
Vitamin D (µg)	1.9 ± 1.1	2.2 ± 1.8	0.434	2.1 ± 3.4	2.3 ± 1.2	0.217	0.931	0.367	0.210
Vitamin E (mg)	13.9 ± 7.8	12.9 ± 6.2	0.681	16.1 ± 7.6	12.4 ± 3.8	**0.019**	**0.047**	**0.018**	0.139
Phenylalanine (mg)	3837.4 ± 1519.3	4033.1 ± 1891.6	0.597	3036.6 ± 1514.6	4850.6 ± 1940.3	**<0.001**	0.461	**0.042**	**<0.001**

**Table 6 nutrients-11-02815-t006:** Concentration of fatty acids measured in blood serum between vegan group and meat-rich group before and after four weeks of intervention ^+^ It was only possible to measure this in 14 VD subjects and 23 MD subjects; VD = Vegan diet; MD = Meat-rich diet; SD = Standard deviation; *p*-value (baselines) indicates comparison of values at baseline (results from *t*-test or MWU-test). *p*-value (end) indicates comparisons at the end of the study adjusted for baseline values (results from ANCOVA). *p*-value (baseline-to-end) indicates comparisons of changes from baseline to end of study (results from mixed ANOVA). Statistically significant results are bolded (*p* < 0.05).

	Baseline			End			*p*-Value (Baseline-to-End)		
	Vegan ± SD (n = 26)	Meat-Rich ± SD (n = 27)	*p*-Value (Baselines)	Vegan ± SD (n = 26)	Meat-Rich ± SD (n = 27)	*p*-Value (End)	Time	Diet	Time × Diet
Arachidonic acid (µM)	4.3 ± 2.0	3.9 ± 1.7	0.790	3.3 ± 1.3	6.4 ± 8.0	**0.046**	0.111	0.080	**0.043**
Docosahexaenoic acid (µM) ^+^	0.2 ± 0.2	0.1 ± 0.1	0.683	0.1 ± 0.1	0.3 ± 0.9	0.298	0.423	0.408	0.263
Eicosanoic acid (µM)	14.9 ± 6.4	11.2 ± 3.5	0.288	12.9 ± 5.7	14.2 ± 11.0	0.211	0.876	0.991	0.124
Eicosenoic acid (µM)	1.1 ± 0.4	0.8 ± 0.4	0.410	0.9 ± 0.3	0.8 ± 0.4	0.906	0.608	0.314	0.387
Linoleic acid (µM)	406.6 ± 179.3	484.2 ± 235.3	0.421	440.3 ± 193.3	401.5 ± 176.3	0.532	0.228	0.588	0.271
Linolenic acid (µM)	3.5 ± 1.4	2.3 ± 0.8	0.742	2.9 ± 1.3	2.3 ± 1.3	0.705	0.061	0.406	0.933
Oleic acid (µM)	97.0 ± 40.4	100.3 ± 48.7	1.000	92.7 ± 51.13	100.5 ± 59.7	0.602	0.338	0.714	0.648

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
