# Peer review of "Vitamin B12 Status Upon Short-Term Intervention with a Vegan Diet—A Randomized Controlled Trial in Healthy Participants"

_nutrients, 2019, doi:10.3390/nu11112815_

Round 1

Reviewer 1 Report

The revised manuscript reads well and previous concerns have been addressed

However, a figure has been added to the discussion in this version. If included in the manuscript, this figure should really be in the introduction section, not the discussion.  

Amendments:

Line 535- the sentence starts with a lower-case letter- correct to upper case.

There are still inconsistent abbreviations of homocysteine in the manuscript – in some places Hcy and in other tHcy. Correct this so the abbreviation is consistent throughout.

Author Response

Thank you again for your careful review of our manuscript! Our response to your concerns is again italicized.

However, a figure has been added to the discussion in this version. If included in the manuscript, this figure should really be in the introduction section, not the discussion.
We moved it to "Introduction".

Line 535- the sentence starts with a lower-case letter- correct to upper case.
Corrected.

There are still inconsistent abbreviations of homocysteine in the manuscript – in some places Hcy and in other tHcy. Correct this so the abbreviation is consistent throughout.
Thank you. We now added tCys and tHcy to the list of abbreviations at the end of the manuscript. tHcy and tCys are standard nomenclatures used in diagnostics to denote total plasma homocysteine and total plasma cysteine concentrations. Hcy and Cys are the standard, generic abbreviations for metabolites homocysteine and cysteine, respectively. We use both abbreviations depending in context. We have now corrected the abbreviations that were not used properly in the previous version of the manuscript.

Reviewer 2 Report

In this reviewers opinion the manuscript is very good as is.

Author Response

Thank you! We are very happy that we were able to meet your concerns!

Reviewer 3 Report

The paper is much improved and clearer.  In particular, the flow is now clear.

L 233: Although 53, and 61 initially consented, were in the study (first paragraph of Results), in the power section you say that you aimed to recruit 48 plus 4 as insurance against dropout.  I suppose you could say that you intended to recruit at least 4 extras.  Please make the text consistent.

The paper is still unusually long among papers published these days.  This is not intrinsically wrong, but will impede readership.  On the other hand, the paper is comprehensive.  Findings are discussed twice, once in Results, the second time in Discussion, and this adds length.  This approach is consistent with the standard approach in this kind of article to keep Results and Discussion separate, of course.  This comment does not require revision, if the editor does not ask you to find a way to make the paper more accessible.

Author Response

Thank you again for your careful review! Our response is again italicized.

The paper is much improved and clearer. In particular, the flow is now clear.

L 233: Although 53, and 61 initially consented, were in the study (first paragraph of Results), in the power section you say that you aimed to recruit 48 plus 4 as insurance against dropout. I suppose you could say that you intended to recruit at least 4 extras. Please make the text consistent.
We added "at least" as you suggested to make it more clear.

This manuscript is a resubmission of an earlier submission. The following is a list of the peer review reports and author responses from that submission.

Round 1

Reviewer 1 Report

There are still several concerns which were raised by this reviewer in the initial review, not visible here with a resubmission.

The major concern is the presentation of the statistical analysis. It seems as though the authors have taken into account the feedback and updated their analysis to the ANCOVA, but still have not presented their data in the order of main effects i.e. is there a group x time interaction, is there an effect of group (i.e. diet) and is there an effect of time (i.e. pre to post). If there are any of these three main effects, then the comparisons can be further probed for where those effects lie (i.e. differences between groups at different timepoints, using the Bonferroni post-hoc tests).

Further, with the comment that was left on the paper by the authors it is clear that the authors do not agree on what their statistical methods were. Why were t-tests still included, when as “LH1” pointed out, the ANCOVA allows for all of the between and within subject comparisons? There is significant bias in the analysis of this dataset as the authors have only tested what is interesting to them and not followed correct statistical procedures.

The authors are still overstating results that are not significant and therefore the results need to be rewritten consistently (i.e. lines 266, 346-347

For the background and therefore hypothesis, where is the information leading to the hypothesis of a quick response of these markers of Vit B12 status? Also, the aim here (lines 86-87) does not match the “major questions” on lines 153-156?

The presentation of data is still not consistent, which was commented in the previous review and not acted upon. Figure 3 has baseline vs post on the x axis and Figure 2 has baseline vs post on the legend – this is confusing for the reader. Also, what data is presented in the box plot? Is the centre of the box the mean or the median? Is the top/bottom of the box 75/25% and the 95% CI as the error bars?

The primary outcome of the study is discussed in lines 556-562, whereas the initial paragraph of the discussion does not present any primary outcome or primary discussion.

The authors have added the CONSORT flow diagram but have not specified how many finished in each group and how many from each group had their dietary intake (“nutritional protocols”) analysed. Please edit.

Why were the 9 participants, who did not follow the dietary protocol or have an adequate dietary intake, included in the biochemical analyses? How do these participants accurately reflect the trial group they were allocated to?

Minor grammatical/clarification issues:

Line 114: how were participants to keep their weight stable? Did this require weekly weighing or some other methodology?

Line 248-251: these results should be presented as mean ± SD for each group, not the two groups combined, to clarify to the reader that they are parallel groups. A table would be the easiest way to do this.

Line 263-264: Were these changes significant? If so, add the statistical comparison to the figure so it is clear and present the mean ± SD of the change.

Line 266: It holo-TC did not “decrease slightly” as the data do not look different and any statistical analysis would say this. Please correct and do not overstate findings. Why are these data being presented twice, i.e. the same in the text as in the table?

Figure 3 and lines 271-273: why are both groups combined here for this analysis?

Lines 281-282: is this increase significant? The SD’s are large and no statistical analyses are presented. Same with lines 285-286, and 290-291?

Line 299: what are “meat-rich subjects”? Do the authors mean the type of diet the subjects were consuming? If so, please rephrase.

In all tables, main effects should be presented (effect of time, group and interaction).

Lines 342-343: is this a result?

Line 369 etc: it does not need to be repeated that it was an ANCOVA analysis, this is stated in the statistical analysis section.

Line 391: this should read “dietary intake of 44 participants (22 VD, 22 MD) were analysed.

Line 405: this should read “during the trial”

Line 405: reword “Besides vitamin B12 nutritional protocols revealed…”

Line 409: Iodine intake was deficient

Line 414-416: There is no difference, so this does not make sense?

Line 422: were not are

Line 426-427: This difference was significant at baseline but is not presented as such.

Line 430: the concentration of what?

Line 431: is this main effect of diet for creatinine or PGEM?

Line 572: energy not caloric; calorie is a unit of energy.

Line 484-488: this is a one sentence paragraph that should be joined with line 489.

Lines 511-515: these references seem to squash the premise of this study by showing that Vit B12 status takes a long time to change. Can the authors then justify why the short time frame was investigated?

Lines 587 and 588: What is “This” that is repeated here?

Reviewer 2 Report

Previous feedback suggestions have been addressed. However, the statistical methods section now reads like a justification to address reviewer feedback. This section needs editing to a more streamlined explanation. 

The manuscript still needs minor editing corrections.  

Reviewer 3 Report

The article is improved by the revision, apart from a few minor problems with syntax.

Line 105- "a free non-recurring breakfast.." maybe replace with "and a single free breakfast was offered".

Line 144-"depended baseline"; omit the word depended

Line 348- "There was not statistically significant" replace"not" with "no"

Line 503- "occurred independently to assigned diet" replace with "independent of the assigned diet"

Line 505-"  end of eth trial not being suggestible by assigned diet" replace perhaps with "not related to the assigned diet"

Line 512- "the quality of liver tissue" do you mena "quality of liver function"?

Line 559= remove the word "down" or replace "drop down" with the word fall.

Reviewer 4 Report

To the Authors:

This report is of a study in 53 people (all meat eaters prior to the study) randomized to eat a vegan or a meat diet for 4 weeks.  The focus is on the conundrum of vegetarian diets being healthier in the long run, but apparently leading to at least a risk of vitamin B12 deficiency.  The study asks what happens in 4 weeks after new initiation of a vegan diet.  This focus is not what is stated in several places, however, and in general the focus seems to wander widely.

The paper is difficult to read.  There are many unusual English phrases.  Results, methods, and discussion are intermixed.  Many nonsignificant findings are emphasized; though mostly reasonable in direction, these may be nothing but noise.  Except under unusual circumstances, it is adequate to report such findings in tables, with brief mention in text.  The paper would benefit in clarity from being much more tightly written.  The paper is substantially too long for easy reading, and much could be said more succinctly.

The Discussion is interesting and well written (though also a lot of words).  The paragraph on holo-transcobalomin (L 510-533) is good, but seems to argue around a false premise.  Holo-TC is a marker that responded rapidly in this study to cessation of meat and dairy, so would seem to indicate a B12 problem, but did not reflect any clinical issues with B12 deficiency over a long run (reference 15 about Indian unsupplemented lifelong vegetarians, aged 29).  You have studied and built up holo-TC here in this study, but now knock it down.  I agree that others may feel that holo-TC is a better marker than circulating B12, and you are displaying the whole picture here.  Still, the presentation seems to be somewhat circular.

I am confused by your language regarding cellular levels of various markers.  You didn’t measure any cellular levels.  I can see that you intend to talk about serum markers which better reflect cellular levels of different compounds.  This assertion may be true, but it is not certain which serum markers do represent cellular contents well.  Quite often serum markers do not mark cellular levels, in my experience.

You also talk a lot about systemic inflammation, but I don’t see that you have measured systemic inflammation?

L 99: Rather than “how to perform their own balanced mixed diet” I think you mean “how to prepare their own balanced mixed diet”

L 144: “depended” should be “dependent”.

L 153-159: It is confusing to talk about 2 primary aims, then say the primary target was the second aim.  This issue recurs in the first paragraph, Lines 450-455.

L 178: You say that these markers are cellular, but they were measured in blood?

Figure 1: The analysis is not intention to treat due to loss of 9 people to noncompliance, therefore baseline balance due to randomization is compromised.  L 241 contradicts the figure by saying all 53 completed the protocol.

L 250: “(after intervention VD mean 22.9 to 22.7 kg/m2 vs. MD mean 23.3 to 23.4 kg/m2, p = 0.240)”  I don’t understand what the cited ranges mean.

L 256: “The effect of diet, measured by ANCOVA, was statistically significant (p  <0.001).” This (and all other aspects in the randomized trial) should be referring to the net difference, change in VD – change in MD.  Please clarify.

L 272: “Figure 272 3 depicts the linear fit of correlation analysis and data dispersion lines.”  What is a data dispersion line?

L 319: What do you mean by “conspicuous values”?  Perhaps “extreme”?